# Metastatic Colorectal Cancer. First Line Therapy for Unresectable Disease

**DOI:** 10.3390/jcm9123889

**Published:** 2020-11-30

**Authors:** Jorge Aparicio, Francis Esposito, Sara Serrano, Esther Falco, Pilar Escudero, Ana Ruiz-Casado, Hermini Manzano, Ana Fernandez-Montes

**Affiliations:** 1Department of Medical Oncology, Hospital Universitario y Politecnico La Fe, E-46007 Valencia, Spain; 2Department of Medical Oncology, Hospital Clinic, E-08041 Barcelona, Spain; ESPOSITO@clinic.cat; 3Department of Medical Oncology, Hospital Universitario Sant Joan de Reus, E-43204 Reus, Spain; sarusky_zgz@hotmail.com; 4Department of Medical Oncology, Hospital Son Llatzer, E-07004 Palma de Mallorca, Spain; efalco@hsll.es; 5Department of Medical Oncology, Hospital Clínico Universitario Lozano Blesa, E-50002 Zaragoza, Spain; pescudero@salud.aragon.es; 6Department of Medical Oncology, Hospital Universitario Puerta de Hierro, E-28220 Madrid, Spain; aruiz.hflr@salud.madrid.org; 7Department of Medical Oncology, Hospital Quirón Salud Palmaplanas, E-07004 Palma de Mallorca, Spain; hermini.manzano@quironsalud.es; 8Department of Medical Oncology, Complejo Hospitalario de Orense, E-32001 Orense, Spain; afm1003@hotmail.com

**Keywords:** colorectal cancer, metastatic disease, chemotherapy, targeted agents

## Abstract

Colorectal cancer (CRC) is a commonly diagnosed malignancy. The prognosis of patients with unresectable, metastatic colorectal cancer (mCRC) is dismal and medical treatment is mainly palliative in nature. Although chemotherapy remains the backbone of treatment, the landscape is changing with the understanding of its heterogeneity and molecular biology. First-line therapy relies on a combination of chemotherapy and targeted therapies, according to clinical patient characteristics and tumor molecular profile. Here we review current evidence from randomized clinical trials for using chemotherapy doublets or triplets, and for the addition of bevacizumab or anti-epidermal growth factor receptor (EGFR) agents. Novel therapies developed for small, selected populations are also discussed.

## 1. Introduction

Colorectal cancer (CRC) is a commonly diagnosed malignancy and ranks among the three leading causes of cancer death in developed countries. The most common histology by far is colorectal adenocarcinoma, and we will refer to this subtype in this review. Despite improvements in screening methodologies, approximately 20% of patients are diagnosed with metastatic CRC (mCRC), which carries a 14% 5-year survival rate. The liver is the most common and often the unique site of metastasis. Patients with Stage IV disease can occasionally undergo curative surgical resection of their metastases with curative intent. However, treatment of mCRC is for the most part palliative in nature.

For patients with unresectable mCRC (representing more than 80% of this population), median survival has significantly increased up to 30 months due to advances in multimodal management, modern cytotoxic chemotherapy (CT) and the introduction of targeted agents. Treatment of mCRC requires a multidisciplinary approach because there are multiple therapeutic options and clinical decision-making is complex. Treatment goals are to prolong overall survival, to improve symptoms of disease and to maintain quality of life for as long as possible [1].

Relevant factors for selecting therapy are disease-related characteristics (clinical presentation, tumor burden, resectability and tumor biology), patient-related factors (performance status, age, comorbidity, socioeconomic factors and expectations) and treatment-related factors (toxicity profile and administration schedule). Results of molecular research have demonstrated the need to profile each mCRC for RAS and BRAF mutations. A better understanding of the heterogeneity of mCRC, including primary tumor location (sidedness), microsatellite instability (MSI) status and other clinically actionable tumor aberrations, is reshaping the therapeutic landscape [2].

Currently, first-line treatment relies on a CT combination, often in association with a biologic agent, according to the individual molecular status. Here we review current evidence from randomized clinical trials (RCTs) for using CT doublets or triplets, and for the addition of bevacizumab or anti-epidermal growth factor receptor (EGFR) agents, in the first-line treatment of patients with mCRC. Novel therapies developed for small, selected populations are also discussed.

## 2. Chemotherapy

Systemic combination CT represents the keystone of treatment for mCRC. Its use improves patient survival and quality of life in comparison with supportive care. Early introduction of treatment has been shown to be superior to treatment delay until the onset of symptoms. Fluoropyrimidines (FPs) constitute the backbone of combination schemes. There are mainly two alternatives, namely 5-fluorouracil (5-FU), administered intravenously, and capecitabine. 5-FU should be used by continuous infusion (CI) because it is more effective and less toxic than bolus administration but requires the implantation of a central venous access device. Capecitabine presents a similar clinical activity and tolerance, with the advantage of its oral administration. RCTs have shown comparable efficacy for both agents, either in monotherapy or in combination. As single agents, they offer response rates (RRs) of about 20% and median overall survival (OS) of 12 months. Toxicity profiles are somewhat different as CI 5-FU induces neutropenia and thromboembolic events more frequently, whereas capecitabine use is associated with a higher incidence of diarrhea and hand-foot syndrome.

By 2000, two new active agents in mCRC had been introduced, namely oxaliplatin (a platinum derivate) and irinotecan (a camptothecin inhibiting topoisomerase I). Their association with either 5-FU or capecitabine significantly improves all parameters of treatment efficacy. Four RCTs have shown that first-line oxaliplatin- or irinotecan-based combination CT schedules offer a similar RR of 34–55%, time to progression (TTP) of 7–8 months and median OS of 14–21 months [3]. Thus, they can be used indistinctly in this clinical setting. Oxaliplatin induces more neurotoxicity and thrombocytopenia, while irinotecan use is more commonly associated with diarrhea and leukocytopenia.

The proportion of patients exposed to all active drugs (FPs, oxaliplatin and irinotecan), either simultaneously or sequentially, strongly correlates with median survival in published large Phase III trials. The likelihood of receiving them is lower with the sequential single-agent treatment strategy than when using a sequence of combination therapies [4]. The main doublet therapies used in first-line treatment are FOLFIRI (CI 5-FU plus irinotecan), FOLFOX (CI 5-FU plus oxaliplatin) and XELOX (capecitabine plus oxaliplatin). Efficacy parameters appear similar although toxicity profiles differ. XELIRI (capecitabine plus irinotecan) is less commonly used due to a higher incidence of severe diarrhea. The triple therapy with FOLFOXIRI has been compared with FOLFOX or FOLFIRI in three trials; superiority for FOLFOXIRI in terms of RR, PFS and OS was suggested in two of them. However, grade 3 to 4 toxic effects were more common with the triplet [5].

The integration of targeted agents (mainly antiangiogenic drugs and anti-EGFR agents) into CT combination schedules has further improved efficacy parameters and survival outcomes, as discussed below. Table 1 shows a summary of the main RCTs of first-line CT schedules for patients with unresectable mCRC evaluating CT doublets vs. triplets and investigating the addition of either bevacizumab or anti-EGFRs [6,7,8,9,10,11,12,13,14,15,16,17,18,19,20,21,22,23,24,25,26,27]. They have been categorized by the quality of design and the achievement of study objectives. The ESMO-Magnitude of Clinical Benefit Scale version 1.1 [28] was also used to assess them.

## 3. Addition of Bevacizumab to Chemotherapy

Patients with RAS mutations represent approximately 50% of the mCRC population. They have fewer treatment alternatives than RAS wild-type patients and also display a worse prognosis (hazard ratio of 1.5–2 for OS). Furthermore, tumor sidedness per se has limited influence on treatment decision in this setting. Multiple clinical trials have shown that tumors with any activating mutation of KRAS or NRAS (exons 2–4) do not benefit from anti-EGFR therapies. Their use can even be detrimental, and so they are contraindicated.

In contrast, bevacizumab, a monoclonal antibody directed against the circulating vascular endothelial growth factor A (VEGF-A), has been authorized since 2004 in combination with CT for the first-line treatment of patients with mCRC, independently of the RAS mutational status. Its toxicity profile is well known, predictable and manageable. Its main adverse events are arterial hypertension, proteinuria, arterial thrombotic events, bowel perforation (uncommon), bleeding and wound healing problems. Bevacizumab induces a consistent improvement in PFS with any of the CT schemes (monotherapy or multiagent) employed in first-line treatment, so its use is standard for most patients with no formal contraindication. The combination of bevacizumab with CT doublets induces objective responses of around 45%, median PFS of 9–10 months and median OS of nearly 24 months. However, the relative added efficacy in terms of survival increment is greater when bevacizumab is combined with less active CT schedules or FP monotherapy.

The addition of the antiangiogenic agent bevacizumab has been evaluated in nine comparative clinical trials [6,7,8,9,10,11,12,13,14] (Table 1), and three of them achieved their prespecified objective. Quality of trial design was quite variable. Significant benefit in survival was seen in only two of them, both using the bolus 5-FU IFL (irinotecan, fluorouracil and leucovorin) regimen, which is considered obsolete nowadays.

Bevacizumab provides a significant incremental gain in PFS when it is added to CT regimens such as 5-FU/leucovorin or bolus IFL, but this gain seems to be reduced when it is combined with more active CT schedules like FOLFOX, XELOX or FOLFIRI. Two trials assessed the combination of bevacizumab with either FOLFOXIRI or FOLFOX and found increased RR. There is no predictive biomarker of bevacizumab efficacy, and patients with either wild-type or mutant RAS may benefit from this therapy.

Recommendation:

In the RAS mutated population, standard first-line treatment consists of a CT doublet, generally associated with bevacizumab, if no contraindications are present.

## 4. Addition of Anti-EGFR Agents to Chemotherapy

Today, determination of RAS and BRAF mutational status is preceptive in order to select the most appropriate treatment for fit patients with unresectable mCRC. RAS mutations are negative predictors of the efficacy of anti-EGFR agents and should always be ruled out before their use. BRAF mutation (e.g., V600E) is less common and confers a poor prognosis. Approximately 40% of patients are RAS and BRAF wild type.

RAS mutation status allows the selection of individuals who might benefit from strategies targeting the EGFR [29]. The anti-EGFR monoclonal antibodies cetuximab and panitumumab should only then be prescribed for patients whose tumors are RAS wild type (KRAS exons 2, 3 and 4 and NRAS exons 2, 3 and 4). The addition of cetuximab or panitumumab to either FOLFIRI or FOLFOX significantly increases patient OS from 20 to 26–28 months [15,18]. However, it is also associated with increased grade 3–4 toxic events (mainly acneiform rash, infusional reactions, diarrhea and hypomagnesemia). Direct comparative studies of cetuximab vs. panitumumab in combination with CT doublets containing CI 5-FU have not shown efficacy differences between both monoclonal antibodies; nowadays, they are considered equivalent. In contrast, the association of anti-EGFR antibodies with either capecitabine or bolus 5-FU plus oxaliplatin is not recommended due to suboptimal results [16,17].

A total of six clinical trials have evaluated the role of anti-EGFR agents in first-line therapy [15,16,17,18,19,20] (Table 1). Most of them were planned for the overall mCRC population and then reanalyzed for RAS wild-type patients. None of them achieved the prespecified objective. Quality of trial design was variable. Significant benefit in survival was observed in half of them. RR and PFS were significantly increased in RAS wild-type patients, whereas the opposite was seen in those with any RAS mutation (negative predictive value).

Recommendation:

CT doublets with anti-EGFR are recommended for fit patients with unresectable mCRC and RAS wild-type tumors.

## 5. Bevacizumab vs. Anti-EGFRs

The addition of both bevacizumab and anti-EGFR agents to combination CT has been shown to be detrimental and should not be used in this setting [30,31].

Data from three head-to-head first-line trials evaluating the activity of anti-EGFR versus bevacizumab in combination with CT doublets for RAS wild-type patients have been reported [21,22,23] (Table 1). Results were discordant and thus nonconclusive. None of them achieved the prespecified end-point, and PFS was equivalent in all three. However, increased RR and a trend toward better survival were observed with anti-EGFR.

Primary tumor location may influence the treatment selection. Results from a joint analysis of trials with anti-EGFR antibodies plus CT showed a greater benefit in left-sided tumors, while greater benefit was observed for right-sided cancers when CT was given alone or combined with bevacizumab [32]. In RAS mutated tumors, first-line treatment should include bevacizumab plus CT, independently of primary site.

Recommendation:

For left-sided, RAS wild-type tumors, the combination of CT plus anti-EGFR might be considered elective.

## 6. Doublets vs. Triplets

5-FU, oxaliplatin and irinotecan are used step by step through two-drug regimens (FOLFOX/XELOX, FOLFIRI/XELIRI), with or without monoclonal antibodies. However, not all patients receive all three of these drugs because their condition may not allow second-line treatment. The optimal sequencing has not yet been established, and some Phase III trials had shown that upfront treatment with 5-FU, oxaliplatin and irinotecan with or without bevacizumab (FOLFOXIRI regimen) can improve outcomes for patients with mCRC [14,24,25,26,27] (Table 1).

In a recent meta-analysis of eight RCTs, FOLFOXIRI was associated with improvements in efficacy outcomes, notably with a 25% survival increase. FOLFOXIRI was also associated with increased grade 3–4 toxicity [5]. A propensity score-adjusted method was conducted to provide an estimation of the benefit from the addition of bevacizumab to FOLFOXIRI. In the absence of a randomized comparison, the addition of bevacizumab to FOLFOXIRI provided significant benefit in PFS and OS, supporting its use as upfront treatment for mCRC patients [33].

Less evidence exists about the treatment with anti-EGFR and triplet therapy. A randomized Phase II trial was recently presented, comparing mFOLFOXIRI plus panitumumab versus FOLFOXIRI in 96 RAS WT patients. ORR and secondary resections of metastases were increased, PFS was similar and OS showed a trend in favor of the panitumumab-containing arm [34].

TRIBE-2 compared a preplanned strategy of upfront FOLFOXIRI followed by the reintroduction of the same regimen after disease progression versus a sequence of mFOLFOX6 and FOLFIRI doublets, in combination with bevacizumab. This Phase III trial showed that upfront FOLFOXIRI plus bevacizumab seems to be a preferable therapeutic strategy in terms of PFS and OS [27].

In most clinical trials, the benefits of triplets are more evident for fit, younger patients with no comorbidity and no prior exposition to oxaliplatin in the adjuvant setting. Based on limited individual patient data, several guidelines recommended FOLFOXIRI plus bevacizumab for patients with BRAF mutation. However, a recent meta-analysis has questioned this recommendation [33].

Recommendation:

FOLFOXIRI with or without bevacizumab is associated with improvements in efficacy outcomes (OS, PFS, RR) in mCRC at the expense of increased toxicity.

Table 2 summarizes current recommendations for first-line treatment according to the individual molecular profile and tumor sidedness.

## 7. Treatment of Other Molecularly Selected Populations

In recent years, several clinical trials have demonstrated that small, molecularly selected subgroups of patients with mCRC may benefit from new directed therapies. Patients with MSI-high or mismatch repair deficient (dMMR) tumors represent less than 5% of mCRC. In these cases, treatment with immunotherapy is more effective than conventional CT (with or without antiangiogenic or anti-EGFR agents), both in first [35] and successive lines of therapy [36]. V600E BRAF mutations are present in 8–15% of mCRC cases and predict a worse prognosis as well as resistance to standard combinations. The association of encorafenib and cetuximab with or without binimetinib has shown promising results in second and later lines of treatment [37] and is now being evaluated in patients receiving less pretreatment. Finally, less than 5% of the mCRC population shows HER2 amplifications (more commonly associated with rectal primaries) or NTRK fusions (more frequently in MSI-high or dMMR cases). These rare alterations are highly actionable with drugs such as trastuzumab [38] or larotrectinib [39], respectively. Nowadays, only immunotherapy with pembrolizumab has been approved for first-line treatment in patients with mCRC and MSI-high tumors.

## 8. Conclusions and Perspectives

CRC remains a leading cause of cancer death in the Western world. Knowledge of primary tumor location along with specific targetable tumor mutations is beginning to change the outlook for patients with mCRC. Treatment of unresectable, metastatic disease mostly relies on palliative CT (generally doublets or triplets) in combination with targeted agents. Molecular profiling is creating new therapeutic options that give patients hope for increased survival. Testing for RAS and BRAF mutations, as well as MSI status, is now mandatory, as it has direct implications for current therapy. Other actionable tumor aberrations are still under active investigation. This is a rapidly changing landscape of predictive biomarkers for the selection of conventional and/or targeted therapeutic plans that will build on new perspectives in mCRC.

## Figures and Tables

**Table 1 jcm-09-03889-t001:** Main randomized clinical trials in first-line treatment for patients with metastatic colorectal cancer (mCRC).

Author, Journal Year (ref)	N	Treatments	HR PFS/OS < 0.8	Adequate Control Arm	PFS Censored <20% at 2y	Any Change in Primary End- Point or Sample	Achieved Pre- Specified Objective	Quality Design	ESMO/MCBS 1.1
**Bevacizumab vs. no bevacizumab**									
Kabbinavar F, JCO 2003 [6]	104	bev plus FU vs. FU	0.54/0.86. 0.62	no	NA	no	NA	2 of 5	1 of 3
Kabbinavar FF, JCO 2005 [7]	209	bev plus FU vs. FU	0.5/0.79. 0.63	no	yes	no	no	3 of 5	0 of 3
Hurwitz H, NEJM 2004 [8]	813	bev plus IFL vs. IFL	0.54/0.66. 0.95	yes	yes	no	yes	4 of 5	2 of 3
Saltz LB, JCO 2008 [9]	1401	bev plus FOLFOX/CAPOX vs FOLFOX/CAPOX	0.83/0.89.0.93	yes	yes	no	no	3 of 5	0 of 3
Tebbutt NC, JCO 2010 [10]	471	bev plus cap or cap/mit vs. cap or cap/mit	0.61/0.86. 0.71	no	yes	no	yes	4 of 5	0 of 3
Stathopoulos GP, Oncology 2010 [11]	222	bev plus IFL vs. IFL	NA	no	NA	no	NA	1 of 5	0 of 3
Guan ZZ, Chin J Cancer 2011 [12]	214	bev plus IFL vs. IFL	0.44/0.62. 0.71	no	yes	NA	NA	2 of 5	2 of 3
Pasardi A, Ann Oncol 2015 [13]	376	bev plus FOLFIRI or FOLFOX vs FOLFIRI or FOLFOX	0.86/1.13. 0.76	yes	yes	yes	no	3 of 5	0 of 3
Loupakis F, NEJM 2014 [14]	508	bev plus FOLFOXIRI vs. FOLFIRI	0.75/0.79. 0.94	yes	yes	no	yes	4 of 5	0 of 3
**Anti-EGFR vs. no anti-EGFR in RAS WT**									
Van Cutsem E, JCO 2015 [15]	430/1198	cet plus FOLFIRI vs. FOLFIRI	0.56/0.69. 0.81	yes	no	yes	no	1 of 5	3 of 3
Maugham TS, Lancet 2011 [16]	729/1630	cet plus FOLFOX or CAPOX vs FOLFOX or CAPOX	0.96/1.04. 0.92	yes	yes	yes	no	2 of 5	0 of 3
Tveit KM, JCO 2012 [17]	274/571	cet plus FLOX vs. FLOX	1.07/1.14. 0.93	no	yes	yes	no	1 of 5	0 of 3
Douillard JY, NEJM 2013 [18]	512/1183	pani plus FOLFOX vs. FOLFOX	0.72/0.78. 0.92	yes	no	yes	no	1 of 5	2 of 3
Bokemeyer C, Eur J Cancer 2015 [19]	87/297	cet plus FOLFOX vs. FOLFOX	0.53/0.94. 0.56	yes	yes	yes	no	3 of 5	0 of 3
Qin S, JCO 2018 [20]	393	cet plus FOLFOX vs. FOLFOX	0.69/0.76. 0.91	yes	yes	yes	no	2 of 5	1 of 3
**Anti-EGFR vs. bevacizumab in RAS WT**									
Venook AP, JAMA 2017 [21]	474/1137	bev plus FOLFOX or FOLFIRI vs cet plus FOLFOX or FOLFIRI	1.03/0.83. 1.24	yes	NA	yes	NA	1 of 5	1 of 3
Heinemann V, Lancet Onol 2014 [22]	342/592	bev plus FOLFIRI vs. cet plus FOLFIRI	0.93/0.7. 1.32	yes	NA	yes	NA	1 of 5	3 of 3
Schwartzberg LS, JCO 2014 [23]	170/283	bev plus FOLFOX vs. pani plus FOLFOX	0.65/0.62. 1.04	yes	yes	yes	NA	2 of 5	3 of 3
**Triplets vs. doublets**									
Souglakos J, Br J Cancer 2006 [24]	283	FOLFIRI vs. FOLFOXIRI	0.83/NA	no	yes	no	no	2 of 5	0 of 3
Falcone A, JCO 2007 [25]	244	FOLFOXIRI vs. FOLFIRI	0.63/0.7. 0.9	no	yes	no	yes	3 of 5	2 of 3
Loupakis F, NEJM 2014 [14]	508	bev plus FOLFOXIRI vs. FOLFIRI	0.75/0.79. 0.94	yes	yes	no	yes	4 of 5	0 of 3
Sastre J, JCO 2019 [26]	349	Bev plus FOLFOX vs. bev plus FOLFOXIRI	0.64/0.84. 0.76	yes	NA	no	yes	3 of 5	1 of 3
Cremolini C, Lancet Oncol 2020 [27]	679	bev plus FOLFOX the bev plus FOLFIRI vs. bev plus FOLFOXIRI then bev plus FOLFOXIRI	0.79/0.82. 0.96	yes	yes	no	yes	3 of 5	1 of 3

HR: hazard ratio, PFS: progression-free survival, OS: overall survival, NA: not available, ESMO/MCBS 1.1: European Society of Medical Oncology Magnitude of Clinical Benefit Scale version 1.1. [28].

**Table 2 jcm-09-03889-t002:** Current recommendations for first-line treatment according to the individual molecular profile and tumor sidedness.

Primary Tumor Sidedness	Right	Left	Left	Left	Any
**Molecular profile**	Any, MSS	Native RAS and BRAF, MSS	Mutant RAS, MSS	Mutant BRAF, MSS	Any, MSI
**Suggested treatment**	CT doublet or triplet plus bevacizumab	CT doublet plus anti-EGFR	CT doublet plus bevacizumab	CT doublet plus bevacizumab	Immunotherapy (pembrolizumab)

MSS: microsatellite stability; MSI: microsatellite instability; CT: chemotherapy.

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
