# Peer review of "Metastatic Colorectal Cancer. First Line Therapy for Unresectable Disease"

_jcm, 2020, doi:10.3390/jcm9123889_

Round 1

Reviewer 1 Report

Comments

  • Colorectal cancer as a term describes a heterogenous group of malignant tumors. The authors should clarify whether the presented therapeutic strategies to manage unresectable tumors refer to a specific malignant subtype such as colorectal adenocarcinoma which is the predominant form of CRC.
  • An inclusion of a comprehensive table to illustrate the most appropriate treatment plan according to the molecular background of patients would be very helpful.
  • The authors should include more information regarding the recent advances of immunotherapy in CRC as well as its potential towards the clinical management of mCRC.
  • The authors should discuss the potential usefulness of antimetabolites as a standalone therapeutic regimen as well as combined with established therapeutic plans.
  • The authors should discuss about rapidly changing landscape of predictive biomarkers for the selection of conventional and/or targeted therapeutic plans as well as their perspectives in CRC.

Author Response

ANSWERS TO THE REVIEWERS

Thank you very much for yours constructive suggestions. We have included most of them (the others are specifically reviewed in different papers projected for this special monographic issue), and modified the manuscript accordingly.

            REVIEWER 1

  • Colorectal cancer as a term describes a heterogenous group of malignant tumors. The authors should clarify whether the presented therapeutic strategies to manage unresectable tumors refer to a specific malignant subtype such as colorectal adenocarcinoma which is the predominant form of CRC.

We have now specified that we refer to management of colorectal adenocarcinoma in paragraph 1 of the introduction: “By far, the most common histology is colorectal adenocarcinoma and we will refer to this subtype in this review”.

  • An inclusion of a comprehensive table to illustrate the most appropriate treatment plan according to the molecular background of patients would be very helpful.

We have added a new Table 2, cited on lines 196-197: “Table 2 summarizes current recommendations for first line treatment according to the individual molecular profile and tumor sidedness”.  

Primary tumor sidedness

Right

Left

Any

Molecular profile

Any,

MSS

Native RAS and BRAF, MSS

Mutant RAS,

MSS

Mutant BRAF,

MSS

Any,

MSI

Suggested treatment

CT doublet or triplet plus bevacizumab

CT doublet plus anti-EGFR

CT doublet plus bevacizumab

CT doublet plus bevacizumab

Immunotherapy (pembrolizumab)

  • The authors should include more information regarding the recent advances of immunotherapy in CRC as well as its potential towards the clinical management of mCRC.

A specific review of immunotherapy in mCRC is to be published in other paper of this monographic  issue

  • The authors should discuss the potential usefulness of antimetabolites as a standalone therapeutic regimen as well as combined with established therapeutic plans.

Antimetabolites (only fluoropyrimidines are active in this disease), alone or in combination, are discussed in the first 3 paragraphs of point 2. Chemotherapy 

  • The authors should discuss about rapidly changing landscape of predictive biomarkers for the selection of conventional and/or targeted therapeutic plans as well as their perspectives in CRC.

This is included in point 7. Treatment of other molecularly selected population. A new sentence has been added: “Nowadays, only immunotherapy with pembrolizumab has been approved for first-line in patients with mCRC and MSI-high tumors”. Also is commented in 8. Conclusions and perspectives. A new sentence has been added: “This is a rapidly changing landscape of predictive biomarkers for the selection of conventional and/or targeted therapeutic plans that will build on new perspectives in mCRC”. Other chapters in this monograph deal specifically with biomarkers and new agents under investigation.

Reviewer 2 Report

The manuscript summarized the oncological recomendation for chemiotherapy with or without the target therapy in the first line in metastatic colorectal cancer based on randomised clinical trials. The Authors have refered to the current ESMO recomendation. In my opinion it is a pity that the Authors did not write more about the new therapies and the sequence of the therapy as first and second or next line. Especially we do not know much about the first line of colorectal cancer patients with MSI high, HER2 amplification and NTRK fusion, because it is not a huge group of patients. 

The paper contains the results of the most important clinical studies.

Author Response

ANSWERS TO THE REVIEWERS

Thank you very much for yours constructive suggestions. We have included most of them (the others are specifically reviewed in different papers projected for this special monographic issue), and modified the manuscript accordingly.

REVIEWER 2

  • The manuscript summarized the oncological recomendation for chemotherapy with or without the target therapy in the first line in metastatic colorectal cancer based on randomised clinical trials. The Authors have refered to the current ESMO recomendation. In my opinion it is a pity that the Authors did not write more about the new therapies and the sequence of the therapy as first and second or next line.

Second and successive lines of therapy, as well as treatment sequence are discussed in other reviews included in this monograph.

  • Especially we do not know much about the first line of colorectal cancer patients with MSI high, HER2 amplification and NTRK fusion, because it is not a huge group of patients. 

A new sentence has been added at the end of point 7. Treatment of other molecularly selected population: “Nowadays, only immunotherapy with pembrolizumab has been approved for first-line in patients with mCRC and MSI-high tumors”.

  • The paper contains the results of the most important clinical studies.

This manuscript is a resubmission of an earlier submission. The following is a list of the peer review reports and author responses from that submission.